MetaBoot: a machine learning framework of taxonomical biomarker discovery for different microbial communities based on metagenomic data

Wang Xiaojun 1 2
Su Xiaoquan 1 3
Cui Xinping 1 4 xinping.cui@ucr.edu
Ning Kang 1 2 3 ningkang@qibebt.ac.cn
1 Bioinformatics Group of Single Cell Center, Shandong Key Laboratory of Energy Genetics and CAS Key Laboratory of Biofuels, Qingdao Institute of Bioenergy and Bioprocess Technology, Chinese Academy of Sciences , Qingdao, Shandong Province , People’s Republic of China
2 University of Chinese Academy of Sciences , Beijing , People’s Republic of China
3 CUDA Research Center, Qingdao Institute of Bioenergy and Bioprocess Technology, Chinese Academy of Sciences , Qingdao, Shandong Province , People’s Republic of China
4 Department of Statistics, University of California , Riverside, CA , USA
Wang Yong
Electronic publication date: 2015 Jul 7
Publication date: 2015
Volume: 3
Electronic Location ID: e993
Received 2015 Mar 29; Accepted 2015 May 11
Copyright: © 2015 Wang et al.
Copyright year: 2015
Copyright holder: Wang et al.
License: This is an open access article distributed under the terms of the Creative Commons Attribution License, which permits unrestricted use, distribution, reproduction and adaptation in any medium and for any purpose provided that it is properly attributed. For attribution, the original author(s), title, publication source (PeerJ) and either DOI or URL of the article must be cited.
License URL: https://creativecommons.org/licenses/by/4.0/

Keywords: Biomarker, Metagenomic, Machine learning, Bootstrap, mRMR, Taxonomical distribution pattern

Funding: Chinese Academy of Sciences e-Science INFO-115-D01-Z006 Ministry of Science and Technology’s high-tech (863) 2012AA02A707 2014AA21502 NSFC 61103167 31072115 This work was supported by the Chinese Academy of Sciences’ e-Science grant INFO-115-D01-Z006, Ministry of Science and Technology’s high-tech (863) grant 2012AA02A707 and 2014AA21502, NSFC grant 61103167, and NSFC grant 31072115. The funders had no role in study design, data collection and analysis, decision to publish, or preparation of the manuscript.

==============================
As more than 90% of species in a microbial community could not be isolated and cultivated, the metagenomic methods have become one of the most important methods to analyze microbial community as a whole. With the fast accumulation of metagenomic samples and the advance of next-generation sequencing techniques, it is now possible to qualitatively and quantitatively assess all taxa (features) in a microbial community. A set of taxa with presence/absence or their different abundances could potentially be used as taxonomical biomarkers for identification of the corresponding microbial community’s phenotype. Though there exist some bioinformatics methods for metagenomic biomarker discovery, current methods are not robust, accurate and fast enough at selection of non-redundant biomarkers for prediction of microbial community’s phenotype. In this study, we have proposed a novel method, MetaBoot, that combines the techniques of mRMR (minimal redundancy maximal relevance) and bootstrapping, for discover of non-redundant biomarkers for microbial communities through mining of metagenomic data. MetaBoot has been tested and compared with other methods on well-designed simulated datasets considering normal and gamma distribution as well as publicly available metagenomic datasets. Results have shown that MetaBoot was robust across datasets of varied complexity and taxonomical distribution patterns and could also select discriminative biomarkers with quite high accuracy and biological consistency. Thus, MetaBoot is suitable for robustly and accurately discover taxonomical biomarkers for different microbial communities.

Introduction

The approximate estimation of microbial cells on earth is 1030 (Proctor, 1994), which is huge, and a large number of novel genes with useful functions might be contained within the genomes of these unknown communities of microbes. However, it was estimated that more than 90% of species in the microbial communities are unknown and uncultivable (Jurkowski, Reid & Labov, 2007). Therefore, the traditional processes for isolation and cultivation of microbes are not applicable for the analyses of many microbial communities. Based on the development of Next Generation Sequencing (NGS), the metagenomic method become one of the important methods that could provide direct access to genomes of as-yet-uncultivated microorganisms in native environments (Eisen, 2007). Metagenomics makes it possible to better understand microbial diversity as well as their functions. Metagenomics has become an increasingly popular research area when its diverse and multiplicity of metagenomics and its potential applications in environmental sciences, bioenergy and human health is considered.

One of the most broadly applicable and successful means of translating molecular and genomic data into applications such as clinical practice (Segata et al., 2011) and environmental monitoring (Lam & Gray, 2003) is the identification of biomarkers. Comparisons among different types of tissues or samples have highlighted the importance of detecting novel subtypes of a disease or determining the subtype of a new sample (Golub et al., 1999; Tothill et al., 2008). In any genomic dataset, identifying the most biologically informative features which can differentiate two or more sets of samples remains an obstacle, and for metagenomic biomarkers this is particularly true.

Other than the challenges associated with high-dimensional data which includes different meta data or data type, metagenomic analysis additionally presented their own specific issues, including sequencing errors, chimeric reads (Swan et al., 2002; Wooley & Ye, 2010) and complex underlying biology (multiple species and their uniqueness, relative abundances, complex functions, etc.). Remarkable inter-subject variability would usually present a profound property of many microbial communities as well, which has made biomarker identification a big hurdle. For instance, both environmental and human microbiomes might be subjected to a long tail distribution of rare organisms (Liao et al., 2011; Pedrós-Alió, 2006). Therefore, robust and efficient bioinformatics tools that could ensure the reproducibility of biomarker identification from metagenomic data, which is crucial for its applications, are needed. Further, as mentioned in Segata et al. (2011), elucidating the biological consistency and roles of selected biomarker, especially non-redundant biomarkers, is a crucial step to understand the underlying mechanisms of community–community or host-community interactions.

A number of methods have been developed for comparison of different metagenomic samples from different angles. Firstly, there are methods that could assess whether communities differ, but not the quantitative assessment of differences and what make the differences. DOTUR (Schloss & Handelsman, 2005) and SONS (Schloss & Handelsman, 2006a) cluster sequences into operational taxonomic units (OTUs) and, by estimating the diversity of a microbial population, provide a coarse measure for comparing different communities. TreeClimber (Schloss & Handelsman, 2006b), UniFrac (Lozupone & Knight, 2005) and Meta-Storms (Su, Xu & Ning, 2012) compare sets of metagenomics in a phylogenetic context. Secondly, there are tools for comparing two sets of samples. MEGAN (Huson et al., 2007) is a metagenomic analysis tool providing a graphical interface that allows users to compare the taxonomic composition of samples, with additions for phylogenetic comparisons and statistical analyses. MEGAN, however, can only compare single pairs of metagenomic samples, which is also the case with STAMP (Parks & Beiko, 2010). Thirdly, statistical model based methods were developed for the comparison of samples. MG-RAST (Meyer et al., 2008), ShotgunFunctionalizeR (Kristiansson, Hugenholtz & Dalevi, 2009), Mothur (Schloss et al., 2009) and METAREP (Goll et al., 2010) all compare metagenomic samples through standard statistical tests. However, none of these methods directly identify biological features responsible for group relationships (Gower, 1966).

The identification of biomarkers for metagenomic data could illustrate the reason for metagenomic sample differences. There are two general approaches for metagenomic biomarker discovery: bottom-up and top-down. The bottom-up method is the one that tested each taxa and selected ones that would led to the variations between groups. Typical bottom-up methods include Wilcoxon rank-sum test (Wilcoxon) (Bauer, 1972). The top-down method is based on statistical analysis of the overall distribution of taxon in the metagenomic samples. Currently, Metastats (White, Nagarajan & Pop, 2009) and LEfSe (Segata et al., 2011) are the only two available methods that explicitly apply statistical assessment of metagenomic difference for metagenomic biomarker discovery. LEfSe further considered biological relevance, biological consistency and effect size estimation of predicted biomarkers. As pointed out by LEfSe (Segata et al., 2011), to ensure reproducibility of biomarker identification from metagenomic data, robust statistical tools are needed, which is also critical for clinical applications. However, none of the aforementioned two methods have addressed the issue of robustness. In addition, redundancy is a serious issue for metagenomic data analysis, especially for biomarker discovery. Taxonomically, as microbial community is dynamic, it is very common that there exist many similar strains as well as multiple similar mutants of the same strain. However, to maximize the power of biomarkers for clinical diagnostic application, it is desirable to find biomarkers that are both distinguishable and representative. Therefore, biomarkers from the same strain and its mutants or from similar strains are considered as redundant biomarkers since they contain similar genetic and/or clinical information. Note that redundancy in biomarker discovery from gene expression data is less of an issue in that even though two or more genes might be similar, they might play significantly different roles in the biological system (biological importance). Additionally, the evolutionary relationship among similar genes might not be that close enough to treat them as the redundant biomarker.

In this work, we present a top-down strategy, MetaBoot, which uses mRMR (Ding & Peng, 2005) and Bootstrap for feature selection from microbial community samples. Strategically, it is a top-down approach in the sense that it first analyzed the overall structure of the microbial community, and then summarized such property for biomarker identification. The MetaBoot framework is based on taxonomical profiles generated from the microbial community’s 16S rRNA gene sequences. It selects discriminative features as candidate features through bootstrap resampling. This general procedure is simple in principle, yet it is significantly different from previous biomarker discovery methods: the final results would be a set of non-redundant and informative features (genes) selected by mRMR, rather than a complex taxonomy structure or a set of many biologically redundant features. Also, it introduces bootstrap resampling procedure to ensure the robustness and reproducibility.

MetaBoot has been put to the test and compared with other methods on well-designed simulated metagenomic datasets with known biomarkers and realistic taxonomical distribution properties. Results have shown that MetaBoot was robust for biomarker discovery across datasets of varied complexity and taxonomical distribution patterns. On real oral and soil metagenomic datasets, MetaBoot could also select discriminative biomarkers with high specificity and clear biological meaning.

Materials and Methods

Data description

Synthetic datasets

We generated three collections of artificial datasets in order to compare MetaBoot with other methods.

Synthetic dataset S1 (normal dataset)

To demonstrate the ability of our method to select features with lower redundancy compared with LEfSe, Metastats and Wilcoxon, we built synthetic dataset S1 (Fig. 1). Dataset S1 includes 2 classes with three subclasses each, and each subclass has 20 samples. For each sample, there are 10 feature groups (with 10 features in each group) for positive biomarkers and 1 feature group (with 900 features) for negative biomarkers. Therefore, there are 1,000 features and 120 samples in total. For each of the 1,000 features, the values is sampled from a Gaussian normal distribution as described in Fig. 1. Dataset S1 has two properties: first, for positive marker groups, features in class 1 and class 2 have clear difference in mean values, and the between-class differences are larger than between-subclass differences. Secondly, there are feature-to-feature variations within the same feature group due to random distribution function. Nevertheless, features within the same feature groups are considered as redundant features in the dataset.

Figure 1 The structure of synthetic dataset S1 (dataset with normal distributions).

There is a 20(samples)*10(features) matrix in each subclass and positive marker group. And data in each matrix was generated by the normal distribution function (rnorm in R) . More specifically, for group 1–5, the mean parameters for subclass 1, 2, 3 were randomly sampled from the vector (11, 12, 13 and 14); while the mean parameters for subclass 4, 5, 6 were randomly sampled from the vector (17, 18, 19 and 20). Data in group 6–10 were generated in a similar way by using these two vectors reversely. The 900 features in negative marker group all had the same mean value of 15. All features had the same standard deviation (sd) parameters.

In the process of analyzing real data (16S rRNA sequencing data from oral samples), we found that the distribution of many features (taxas) conformed a mixture of normal and gamma distribution or gamma distribution (Fig. 3). For some real data, the defects of measurement could lead to this result. But there is the possibility that features whose distribution conform gamma distribution in real data do exist. Therefore, we built the synthetic dataset S2 (mixture dataset) and S3 (gamma dataset). There are two important parameters, shape and rate, in gamma distribution and both parameters are positive real numbers. Because the change of shape parameter has a greater impact upon the shape of gamma distribution than that of rate parameter, most of the positive markers among subclasses have different shape parameter. The biomarkers that could differentiate “class 1” and “class 2” samples were the subject of biomarker identification.

Synthetic dataset S2 (mixture dataset)

The detailed parameter settings were shown in Table 1. For positive marker groups 1–5, features in class 1 and class 2 have clear difference in shape values. And for positive marker groups 6–10, features in class 1 (gamma distribution) and class 2 (normal distribution) have clear difference in mean values. (The mean and sd values of features in class 2 are determined based on mean and sd values from corresponding features in class 2 with gamma distribution.) Dataset S2 (mixture dataset) has three properties: first, for positive marker groups, features in class 1 and class 2 have clear difference in shape or mean values, and the between-class differences are larger than between-subclass differences. Secondly, for negative marker groups, there is no difference between classes in mean values. Thirdly, there are feature-to-feature variations within the same feature group due to random distribution function. Nevertheless, features within the same feature groups are considered as redundant features in the dataset S2. The biomarkers that could differentiate “class 1” and “class 2” samples were the subject of biomarker identification.

Table 1 The structure of synthetic dataset S2 (dataset with mixture distributions).

In positive marker group, each square is a 25(samples)*10(features) matrix in which each feature was generated by gamma (the red cells) or normal (the green cells) distribution function (generated by rgamma or rnorm in R). But in negative marker group, each square is a 25(samples)*900(features) matrix in which each feature was also generated by normal distribution function.

Class	Sub- class	Positive marker group	Negative marker group	
		1	2	3	4	5	6	7	8	9	10		
Class 1	1	Shape	Shape	Shape	Shape	Shape	Shape	Shape	Shape	Shape	Shape	Mean	
7.18	0.61	1.70	0.81	2.36	7.18	0.61	1.70	0.81	2.36	0.14	
Rate	Rate	Rate	Rate	Rate	Rate	Rate	Rate	Rate	Rate	sd	
44.38	71.12	517	79.70	316	44.38	71.12	517	79.70	316	0.06	
2	Shape	Shape	Shape	Shape	Shape	Shape	Shape	Shape	Shape	Shape	Mean	
6.98	0.51	1.80	0.91	2.46	6.98	0.51	1.80	0.91	2.46	0.14	
Rate	Rate	Rate	Rate	Rate	Rate	Rate	Rate	Rate	Rate	sd	
44.38	71.12	517	79.70	316	44.38	71.12	517	79.70	316	0.06	
Class 2	3	Shape	Shape	Shape	Shape	Shape	Mean	Mean	Mean	Mean	Mean	Mean	
5.70	0.85	1.32	0.33	2.88	0.14	0.009	0.005	0.004	0.009	0.14	
Rate	Rate	Rate	Rate	Rate	sd	sd	sd	sd	sd	sd	
44.38	27.40	210	91.20	507	0.06	0.007	0.002	0.006	0.06	0.06	
4	Shape	Shape	Shape	Shape	Shape	Mean	Mean	Mean	Mean	Mean	Mean	
6.60	0.75	1.22	0.43	2.98	0.13	0.010	0.004	0.003	0.010	0.14	
Rate	Rate	Rate	Rate	Rate	sd	sd	sd	sd	sd	sd	
44.38	27.40	210	91.20	507	0.06	0.007	0.002	0.006	0.06	0.06	

Synthetic dataset S3 (gamma dataset)

The detailed parameter settings were shown in Table 2. Dataset S3 (gamma dataset) has three properties: first, for positive marker groups, features in class 1 and class 2 have clear difference in shape values, and the between-class differences are larger than between-subclass differences. Secondly, for negative marker groups, there is no difference between classes in shape values. Thirdly, there are feature-to-feature variations within in the same feature group due to random function. Nevertheless, features within the same feature groups are considered as redundant features in the dataset S3. The biomarkers that could differentiate “class 1” and “class 2” samples were the subject of biomarker identification.

Table 2 The structure of synthetic dataset S3 (dataset with gamma distributions).

In positive marker group, each square is a 20(samples)*10(features) matrix in which each feature was generated by gamma distribution function (rgamma in R). But in negative marker group, each square is a 20(samples)*300(features) matrix in which each feature was also generated by gamma distribution function.

Class	Sub- class	Positive marker group	Negative marker group	
		1	2	3	4	5	6	7	8	9	10	1	2	3	
Class 1	1	Shape	Shape	Shape	Shape	Shape	Shape	Shape	Shape	Shape	Shape	Shape	Shape	Shape	
7.18	0.61	2.22	1.70	1.29	0.87	0.81	2.56	1.50	1.66	6.20	3.10	0.61	
Rate	Rate	Rate	Rate	Rate	Rate	Rate	Rate	Rate	Rate	Rate	Rate	Rate	
44.38	71.12	33.40	517	94.70	203	79.70	316	44.4	66.16	24.30	66.40	71.10	
2	Shape	Shape	Shape	Shape	Shape	Shape	Shape	Shape	Shape	Shape	Shape	Shape	Shape	
7.38	0.71	2.12	1.80	1.19	0.67	0.91	2.46	1.50	1.56	6.20	3.10	0.61	
Rate	Rate	Rate	Rate	Rate	Rate	Rate	Rate	Rate	Rate	Rate	Rate	Rate	
44.38	71.12	33.40	517	94.70	203	79.70	316	44.4	66.16	24.30	66.40	71.10	
3	Shape	Shape	Shape	Shape	Shape	Shape	Shape	Shape	Shape	Shape	Shape	Shape	Shape	
6.98	0.51	2.02	1.90	1.09	0.77	1.01	2.36	1.50	1.46	6.20	3.10	0.61	
Rate	Rate	Rate	Rate	Rate	Rate	Rate	Rate	Rate	Rate	Rate	Rate	Rate	
44.38	71.12	33.40	517	94.70	203	79.70	316	44.4	66.16	24.30	66.40	71.10	
Class 2	4	Shape	Shape	Shape	Shape	Shape	Shape	Shape	Shape	Shape	Shape	Shape	Shape	Shape	
		5.70	0.85	1.72	0.92	0.50	1.37	0.53	3.28	0.91	2.49	6.20	3.10	0.61	
Rate	Rate	Rate	Rate	Rate	Rate	Rate	Rate	Rate	Rate	Rate	Rate	Rate	
44.38	27.40	37.68	210	66.20	734	91.20	507	42.32	171	24.30	66.40	71.10	
5	Shape	Shape	Shape	Shape	Shape	Shape	Shape	Shape	Shape	Shape	Shape	Shape	Shape	
5.60	0.75	1.62	0.82	0.40	1.47	0.43	3.28	0.81	2.39	6.20	3.10	0.61	
Rate	Rate	Rate	Rate	Rate	Rate	Rate	Rate	Rate	Rate	Rate	Rate	Rate	
44.38	27.40	37.68	210	66.20	734	91.20	507	42.32	171	24.30	66.40	71.10	
6	Shape	Shape	Shape	Shape	Shape	Shape	Shape	Shape	Shape	Shape	Shape	Shape	Shape	
5.80	0.95	1.52	0.72	0.60	1.57	0.33	3.28	0.71	2.59	6.20	3.10	0.61	
Rate	Rate	Rate	Rate	Rate	Rate	Rate	Rate	Rate	Rate	Rate	Rate	Rate	
44.38	27.40	37.68	210	66.20	734	91.20	507	42.32	171	24.30	66.40	71.10	

Real datasets

Oral dataset1: oral samples from Huang et al. (2014)

Supragingival plaques were sampled from fifty volunteers recruited at Day-21, Day 0 (Baseline) and Day-21 (different from the previous Day-21). In this experiment, based on these three time-points, we have generated two groups of samples: (1) The “oral hygiene phase” (Day-21 to Day 0) group, also referred to as EG group. (2) The “experimental gingivitis phase” (Day 0 to Day 21) group, also referred to as NG group. Totally, oral dataset1 includes 100 samples (50 samples for each group). For each of samples, 16S rRNA gene sequencing data were generated, and microbial community structure were then analyzed by Mothur (Schloss et al., 2009) for taxa and their relative abundances in the sample. The biomarkers that could differentiate “oral hygiene phase” and “experimental gingivitis phase” were the subject of biomarker identification.

Oral dataset2

Oral dataset2 from Human Microbiome Project (HMP, http://www.hmpdacc.org) includes 812 samples in which 344 samples are from saliva and other 468 samples are from subgingival plaque. Oral dataset2 includes 44, 69 and 96 features at order, family and genus level, respectively. For each of samples, 16S rRNA sequencing data were generated, and microbial community structure were then analyzed by Parallel-Meta (Su et al., 2014) for taxa and their relative abundances in the sample. The biomarkers that could differentiate “saliva” and “subgingival plaque” origins were the subject of biomarker identification.

Soil dataset: soil samples from Caporaso et al. (2011)

Soil dataset includes 14 samples for which 7 samples were collected from two kinds of soil environment each with different pH values (pH = 4.9 and 8.4, respectively). For each of samples, 16S rRNA sequencing data were generated, and microbial community structure were then analyzed by Parallel-Meta for taxa and their relative abundances in the sample. The biomarkers that could differentiate “pH = 4.9” and “pH = 8.4” were the subject of biomarker identification.

MetaBoot algorithm

The overall MetaBoot algorithm includes (1) normalization step, (2) first feature selection step, (3) bootstrap and feature selection step and (4) feature rank step. Figure 2 is the flow chart of MetaBoot process.

Figure 2 The flow chart of MetaBoot process.

Data normalization

To account for difference of read counts across multiple samples in magnitude, we pre-process the data and convert the raw read counts into relative abundances with per-sample normalization to sum to one (raw read counts/total counts in each sample). And the feature whose 80% values are 0 should be deleted. Notice that for each of samples from real datasets, 16S rRNA sequencing data analyzed by Parallel-Meta (Su, Xu & Ning, 2012) for taxa and their relative abundances in the sample. Every taxa’s relative abundances were already normalized by Parallel-Meta as default setting.

Dataset is discretized before input into mRMR feature selection process. The discretization of the data into categorical data not only helps reduce the substantial noise contained in raw data but also increases the power of mRMR method selecting discriminative features. In our method, we use the method mentioned in previous work (Ding & Peng, 2003) to discretize our data into categorical data. Each feature (also called attribute or variable) of data is discretized using its μ (mean) and σ (standard deviation): any data larger than μ + σ/2 are converted into 1; any data smaller than μ − σ/2 are converted into −1; otherwise, data are converted into 0.

Main process

The input dataset for feature selection are required to be normalized data.

(1) In the first feature selection step, a number of candidate features (Parameter 1, M. M represents the number of features in the first feature selection step.) would be selected by mRMR that could discriminate different samples, but might include many redundant features. Therefore, we employed the following two steps to minimize redundancy. The dataset which included M selected features would be used in the subsequent steps.

(2) The bootstrap process (parameter 2, B. B represents the number of bootstrapping process in this step) is employed to eliminate negative markers and redundant positive markers. Here we have implemented bootstrapping with a principle that the number of samples in each subclass (For example, subclass 1 in Fig. 1; or, alternatively, class when the original data has no subclasses) of the bootstrapped dataset must be equal to that in the same subclass (or class) of original dataset. In other words, we require that the new dataset generated by bootstrapping has the same structure as original dataset. The only difference between original datasets and bootstrapped datasets would be that some samples may appear more than once and some samples may not appear in new dataset.

(3) At the feature rank step, the top N (Parameter 3, M′. M′ represents the number of final features selected) discriminative features from each bootstrapped dataset will be selected by mRMR. All selected features were ranked according to the number of occurrences. M′ of the top ranked features will be selected as our final biomarkers.

The 3 parameters involved in this process could be adjusted according to each project’s requirement and specific metagenomic data. Yet it should be emphasized that M must be greater than M′.

Assessment methods for comparison of different biomarker identification methods

To evaluate and compare different biomarker identification methods, we have defined the redundancy rate, non-redundancy rate, error rate, and classification accuracy as follows: (1) Redundancy rate=# redundancy features# features selected*100%

(2) Non-redundancy rate=1−Redundancy rate

(3) Error rate=# negative features# features selected*100%

(4) Classification accuracy=# samples correctly classified# samples in testing dataset*100%.

Implementation and availability of the method

The MetaBoot method is implemented in MATLAB. The software and simulated data that used in this paper could be found online at http://www.computationalbioenergy.org./metaboot.html. The original mRMR codes are wrapped for feature selection module within MetaBoot. Therefore, MetaBoot cannot be used for commercial application without consent from the author of mRMR and MetaBoot.

The selection standard or parameter setting for different methods

LEfSe: Selecting the features with (1) lower p-value and (2) higher effect size (Segata et al., 2011). About parameter setting, we used the default parameters.

Metastats: Selecting the features with lower p-value (White, Nagarajan & Pop, 2009). About parameter setting, we used the default parameters.

Wilcoxon: Selecting the features with lower p-value.

MetaBoot: Selecting the features with higher bootstrapping frequency.

LIBSVM: optimizing the parameters by using the script (easy.py) to achieve the best classification accuracy. Therefore, for different datasets, the parameters might be different.

mRMR: the feature selection scheme we used was MID (Mutual Information Difference) (Ding & Peng, 2005).

Results and Discussions

One bottleneck for assessment of the effectiveness of MetaBoot for identifying biomarkers from microbial community data is the lack of “ground truth.” To overcome this problem, we have first analyzed taxonomical distribution properties of real metagenomic samples, and generated sets of synthetic datasets with known ground truth biomarkers and distribution properties learned from real data. Secondly, we have analyzed the effects of different parameters on MetaBoot results, using synthetic datasets. Thirdly, we have evaluated the overall performance of MetaBoot by comparing with other methods. Finally, we have assessed the effectiveness of MetaBoot on real datasets.

Taxonomical distribution patterns of real metagenomic samples

One of the most critical problems in identification of biomarkers from microbial community data is the lack of “ground truth.” Although a simulated synthetic dataset could contain such “ground truth,” simulating taxonomical distribution properties of real metagenomic samples is critical for the validity of such synthetic dataset.

In this work, we used oral dataset1 to analyze distribution properties of real metagenomic samples. Also, we have generated 3 sets of synthetic metagenomic datasets. Firstly, some literatures suggested the taxonomical distribution of microbial community conform to normal distribution (Segata et al., 2011). Therefore, we have generated synthetic datasets S1 (Normal dataset) based on normal distributions (see ‘Materials and Methods’ for details).

Secondly, we have evaluated the taxonomical distribution properties for taxa at genus level as features. Based on the analysis of the distribution of oral microbial community dataset (dataset described in “Materials and Methods”), we observed that the distribution of a couple of features (about 10% taxa) conformed a mixture of normal and gamma distribution. For example, taxon Leptotrichia and its mixture of distributions were shown in Figs. 3A–3C. Therefore, we generated synthetic dataset S2 (Mixture dataset) based on the mixture of normal and gamma distribution (see “Materials and Methods” for details).

Figure 3 The distribution plot of taxon Leptotrichia and Actinonyces.

(A) The distribution of relative abundances for taxon Leptotrichia based on all samples in two categories (EG and NG) from Oral dataset1 (refer to “Materials and Methods” for details). The x-axis is relative abundance, and y-axis represents the number of samples. (B) The QQ plot of class EG (the red line in (A)) in taxon Leptotrichia. The p-value of Shapiro–Wilk Normality Test (Shapiro & Wilk, 1965) is 0.93. (C) The QQ plot of class NG (the green line in (A)) in taxon Leptotrichia. The p-value of Shapiro–Wilk Normality Test is 0.02. But the p-value of Kolmogorov–Smirnov Tests (Birnbaum & Tingey, 1951) (KS test) is 0.46 when testing whether the distribution of class NG (the green line in (A)) in taxon Leptotrichia conform gamma distribution. (D) The distribution of EG and NG for taxa Actinonyces. The x-axis is relative abundance, and y-axis represents the number of samples.

Thirdly, we have found that the distribution of over 40% taxa (one example for taxon Actinonyces shown in Fig. 3D) in oral dataset1 conformed gamma distribution tested by the Kolmogorov–Smirnov Tests (Birnbaum & Tingey, 1951) (function ks.test in R). The p-values of KS test were 0.78 and 0.93, respectively, for the two sets (EG and NG) of samples. Therefore, we generated synthetic dataset S3 (Gamma dataset) based on gamma distribution (see “Materials and Methods” for details).

MetaBoot analysis

Here we chose taxa at genus level for analysis, which could be accurately identified by Mothur (Schloss et al., 2009) and Parallel-Meta (Su, Xu & Ning, 2012) software based on the OralCore (Griffen et al., 2011) and GreenGenes (DeSantis et al., 2006) databases, and are detailed enough and widely used for differentiating ingredients of communities. For each synthetic datasets (S1, S2 and S3), we aimed to differentiate “class 1” and “class 2” samples using MetaBoot (see “Materials and Methods” for details).

The MetaBoot process includes 3 major steps: first feature selection step, bootstrap and feature selection step, feature rank step. Throughout the entire workflow of MetaBoot, 3 parameters (M, M′ and B, see “Materials and Methods” for details) are most important for the quality of selected biomarkers.

For synthetic dataset S1, M was set to be 50, because we observed that when M was set to 50, enough or all unique positive features could be obtained from 1,000 features using mRMR (Fig. 4A). Notice that we treated features from the same group as redundant features. After eliminating redundant features, the remaining features were unique features. If the unique features were from positive marker groups, we called those as unique positive features. Since synthetic dataset S1 only includes 10 positive marker groups, we set M′ to be 10. In order to determine parameter B, we set a series gradient of the bootstrap process. We observed that when B was more than 40, the number of total unique features selected did not increase. Therefore, the B value was set to 40 (Fig. 4B). For synthetic dataset S2 and S3, we have observed similar patterns (see Supplemental Information 1 for details). Therefore, in this work, parameters M, B and M′ were set to be 50, 40 and 10, respectively, for all datasets.

Figure 4 Plots for selecting M and B for MetaBoot analysis of synthetic data S1.

(A) The x-axis is the values of M, and the y-axis is the number of unique positive features selected by mRMR for each given M. (B) The x-axis is the number of bootstraps B, and the y-axis is the number of unique features selected by all bootstrap processes. Both (A) and (B) considered different standard deviations (sd) used in synthetic dataset S1.

A comparison with current tools using synthetic data

Redundancy analysis based on synthetic datasets

For comparison of 4 methods as regard to redundancy rate (Eq. (1)), non-redundancy rate (Eq. (2)) and error rate (Eq. (3)), we applied LEfSe, Metastats, a bottom-up method Wilcoxon rank-sum test (Wilcoxon) and our method (MetaBoot) on synthetic dataset S1 (There are 10 positive biomarker groups and each group has 10 redundant biomarkers.), respectively. As shown in Fig. 5, MetaBoot can select more non-redundant positive features than LEfSe, Metastats and Wilcoxon. Additionally, because the 100 positive markers have the same p-value (see “Materials and Methods” for details), Metastats in Fig. 5 does not include error bars which indicate that the 10 selected features are from the same positive marker group (the first positive maker group). Therefore, Metastats could not eliminate redundant features when analyzing synthetic dataset S1.

Figure 5 Comparison of results by 4 methods for synthetic dataset S1 in selecting non-redundant features.

The x-axis is the standard deviation (sd) representing the parameter sds in synthetic dataset S1. The y-axis is the non-redundancy rate Eq. (2) in 10 selected features. The error bar represents 95% confidence interval.

For synthetic dataset S2 (Table 3), MetaBoot could select at least 4 out of 10 non-redundant positive biomarkers which was better than other three methods. For synthetic dataset S3 (Table 3), LEfSe and Metastats could only select less than 5 out of 10 non-redundant positive features on average. Both Wilcoxon and MetaBoot outperformed LEfSe and Metastats in that they both can select at least 5 out of 10 non-redundant positive biomarkers. Among these two, MetaBoot was slightly better than Wilcoxon in selecting non-redundant positive markers.

Table 3 Results about redundancies when applied these methods on synthetic dataset S2 (Mixture dataset) and S3 (Gamma dataset) to select 10 features.

In columns for “LEfSe,” “Metastats,” “Wilcoxon” and “MetaBoot,” the values were the non-redundancy rate (Eq. (2)) of non-redundant biomarkers with standard deviation of 1.

Dataset	LEfSe	Metastats	Wilcoxon	MetaBoot	
S2 (Mixture dataset)	36.0 ± 5.5	26.0 ± 5.5	38.0 ± 8.4	42.0 ± 4.5	
S3 (Gamma dataset)	46.0 ± 11.4	31.4 ± 9.0	50.0 ± 12.2	50.9 ± 8.1	

When we further analyzed the differences between MetaBoot and mRMR, we could observe that MetaBoot had similar ability with mRMR in selecting unique positive markers based on synthetic dataset S1 (see Fig. S2 for details), S2 (non-redundancy rate: 48.0% ± 11.0) and S3 (non-redundancy rate: 53.6% ± 10.1). However, for most synthetic datasets from S1, S2 and S3, mRMR usually had about 10% error rate (Eq. (3)), while MetaBoot had much lower error rate (details of results not shown here).

Robustness analysis based on synthetic datasets

We have applied LEfSe, Metastats, Wilcoxon and MetaBoot on synthetic dataset S1, S2 and S3 to study their robustness defined by their ability to differentiate positive and negative biomarkers, respectively. For each method, 100 features (equal to the number of redundant positive markers in synthetic datasets) were selected as biomarkers; then, the correctly detected biomarkers were counted. Results (Table 4 and Fig. 6) have shown that MetaBoot and Wilcoxon method can detect larger number of correct biomarkers compared to other methods. Although all four methods were shown to be robust on synthetic dataset S1 (based on normal distribution), Wilcoxon and MetaBoot outperformed Metastats and LEfSe greatly on synthetic dataset S2 (based on the mixture of normal and gamma distribution) and S3 (based on gamma distribution), indicating the superiority of Wilcoxon and MetaBoot methods as regard to robustness.

Figure 6 Comparison of results by 4 methods for synthetic dataset S1 in selecting positive features.

The x-axis is the standard deviation (sd) representing the parameter sds in synthetic dataset S1. The y-axis is the number of positive features in 100 selected features. The error bar represents standard deviation of 1.

Table 4 Results about robustness when applied these methods on synthetic dataset S2 (Mixture dataset) and S3 (Gamma dataset) to select 100 positive features.

In columns for “LEfSe,” “Metastats,” “Wilcoxon” and “MetaBoot,” the values were “# of positive features” with standard deviation of 1.

Dataset	LEfSe	Metastats	Wilcoxon	MetaBoot	
S2 (Mixture dataset)	67.2 ± 2.6	48.6 ± 4.0	69.0 ± 2.5	70.1 ± 1.1	
S3 (Gamma dataset)	70.4 ± 5.5	73.3 ± 2.9	83.4 ± 2.3	81.6 ± 2.8	

As regard to robustness, MetaBoot was slightly better than mRMR in selecting positive markers based on synthetic dataset S1 (see Fig. S2 for details), S2 (#positive features: 67.4 ± 3.6) and S3 (#positive features: 80.2 ± 3.0). The built-in bootstrap process in MetaBoot might attribute to MetaBoot’s advantage in selecting more positive biomarkers compared to mRMR.

Classification accuracy analysis based on synthetic datasets

For comparison of different methods in classification accuracy (Eq. (4)), we have applied LEfSe, Metastats, Wilcoxon and MetaBoot on synthetic dataset S3 to select 10 features by each of the methods. We then used these 10 features to perform classification by utilizing Support Vector Machine (SVM) implemented by LIBSVM (Chang & Lin, 2011). The reason that we have not done classification based on synthetic dataset S1 was the large difference between 2 classes, making classification easy-proof by all methods.

Each class has 60 samples in synthetic dataset S3. We have performed 6-fold cross-validation to estimate the classification accuracy. Therefore, in the aforementioned formula, the average classification accuracy is shown in Fig. 7. The highest accuracy was obtained when using 10 features selected by MetaBoot. We also observed that MetaBoot had the most stable classification performance (Fig. 7). We obtained similar results for synthetic dataset S2 (see Fig. S3 for details).

Figure 7 Comparison of accuracies when using 10 features selected by 4 methods based on synthetic dataset S3.

The x-axis represents 4 methods and y-axis represents classification accuracy by SVM.

Biomarker identification based on real metagenomic datasets

Results on oral dataset1

For this dataset, we aim to identify biomarkers that could differentiate “oral hygiene phase” and “experimental gingivitis phase” from 16S rRNA gene sequencing data (details in “Materials and Methods”). We have applied the same four methods on oral dataset1 to select 10 features. Biomarker identification results were shown in Fig. 8.

Figure 8 Biomarker identification results on oral dataset1.

(A) The Venn diagram when we selected 10 features from oral dataset1 using four methods. (B) Circular phylogenetic tree of oral dataset1 at genus level. The tree was generated with RAxML and viewed in ITOL (Letunic & Bork, 2007). Genera are color-coded by phyla, except for the Firmicutes and Proteobacteria, which are shown at class level. We used the same phylogenetic tree plot from microbiome.osu.edu (Griffen et al., 2011), and we added legends onto this tree to show biomarkers selected by different methods.

From Fig. 8A, we observed that MetaBoot selected similar features (9 overlaps) with Wilcoxon, while only 4 and 5 feature overlapping with MetaBoot were found for LEfSe and Metastats, respectively. As shown in Fig. 8B, the 10 features selected by each of these methods could be assigned to 6–7 phyla which are mostly overlapping. As shown in Fig. 8, we observed that Streptococcus were selected by MetaBoot, as well as LEfSe and Metastats. Streptococcus was linked with all kinds of oral problems (Munro & Grap, 2004; Fitzgerald, 1960; Jenkinson & Lamont, 2005). Therefore, Streptococcus can serve as biomarker to distinguish different samples and be used for oral diagnosis (Bisno et al., 1997). Rothia were selected by Wilcoxon, as well as LEfSe. Rothia is part of the normal community of microbes residing in the mouth. Previous work found Rothia in 3% of isolates of nitrate-reducing bacteria from the mouth (Doel et al., 2005).

To compare the discriminations accuracy of 10 features selected by different methods, we performed classification by LIBSVM (Chang & Lin, 2011). Each class in oral dataset1 has 50 samples. And we did 5-fold cross-validation (40 samples are used as training datasets) to estimate the classification accuracy. The classification results were shown in Fig. 9, from which we could observe that MetaBoot still had the highest accuracy and the most stable classification performance.

Figure 9 Comparison of accuracies when using 10 features selected by 4 methods based on oral dataset1.

The x-axis represents 4 methods and the y-axis represents the classification accuracy by SVM.

In order to evaluate the added value of bootstrap on mRMR (as realized in MetaBoot), we have also compared the results of mRMR vs. MetaBoot. Streptococcus (mentioned above) was linked with various oral problems. On the other hand, Cardiobacterium, selected only by mRMR as biomarker, was reported to be a rare cause of endocarditis (Han & Falsen, 2005; Slotnick & Dougherty, 1964), but it was not reported as oral related microbial biomarker in any known studies. The difference between mRMR and MetaBoot can be attributed to the bootstrap process included in MetaBoot. Therefore, apart from advantage in robustness, biomarkers selected by MetaBoot were considered more biologically meaningful comparing to mRMR (see Fig. S4 for details).

Results on oral dataset2

For this dataset, we aim to identify biomarkers that could differentiate “saliva” and “subgingival plaque” from 16S rRNA sequencing data (details in “Materials and Methods”). We have applied the same four methods on oral dataset2 at order, family and genus level to select 10 features, respectively. As shown in vennplot (Fig. 10A), at the level of genus and family, LEfSe, Wilcoxon and MetaBoot had good coherence. Yet at order level, there were larger differences among results from different methods. In addition, considering the complexity of the data, studies of microbial community biomarker at order level would not be as reliable as on genus and family levels and seldom used. Therefore, we only focused on the difference among different methods at the level of genus and family. At the genus level (Fig. 10A genus), Peptostreptococcus (Fig. 10B), which was selected by MetaBoot, has been isolated from a wide range of human oral infections (Downes & Wade, 2006) and implicated in human gingivitis and periodontitis (Riggio & Lennon, 2003). At family level (Fig. 10A family), Spirochaetaceae (Fig. 10B) was selected by MetaBoot but not other method. It was also interesting to observe that all oral spirochetes (belonging to Spirochaetaceae family) were classified in the genus Treponema (Chan & McLaughlin, 2000), and Treponema was reported to be associated with periodontal diseases (Chan & McLaughlin, 2000; Sela, 2001). But for Propionibacteriaceae (Fig. 10B), which was selected by LEfSe and Wilcoxon, though this species could be isolated from normal, gingivitis and periodontitis sample with small amount (Riggio et al., 2011), there was few report about the relationship between oral disease and Propionibacteriaceae. Therefore, these results on real oral samples have clearly shown the advantage of MetaBoot on discovery of biologically meaningful biomarkers.

Figure 10 Biomarker identification results on oral dataset2.

(A) The Venn diagram when 10 features were selected at different level from oral dataset2 using the methods. (B) The bar-chart of average relative abundance of the features selected by MetaBoot or LEfSe and Wilcoxon.

Results on soil samples

For this dataset, we aim to identify biomarkers that could differentiate “pH = 4.9” and “pH = 8.4” from 16S rRNA sequencing data (details in “Materials and Methods”). Unlike two previous oral datasets that we have used in “Results on oral dataset1” and “Results on oral dataset2,” each class in soil dataset only has 7 samples. Therefore, we focused on the different features selected by different methods not the distribution properties of features. (The sample size is small for distribution analysis). Due to the complexity of soil microbial community samples, we chose taxa at phylum level for analysis.

When we performed classification by LIBSVM (Chang & Lin, 2011), the classification accuracy was always 100% regardless of either of the 5 or 10 features (selected by the four different methods) we used. For soil dataset, features selected by the four different methods all had distinguishing ability to identify different samples. However, biological explanation of features selected by the four different methods needed further research.

Based on the above results for soil samples, we could observe that features selected by the four different methods were quite different (Fig. 11A), yet most of these features had distinguishing power to identify different samples. Further investigation and interpretation of these features might provide more biological insights for the underline functionality of microbial community.

Figure 11 Biomarker identification results on soil dataset.

(A) The Venn diagram when we selected top 10 features from soil dataset using the four methods. (B) The bar-chart of average relative abundance of 5 features selected by MetaBoot under different pH values. The values for “Others” are computed as the average for other taxa. The dataset is small for standard parametric approaches. Therefore, the p-values (*, 0.01 ≤ p-value < 0.05; **, p-value < 0.01) were calculated through permutation tests (a one-way exact test) (Kabacof, 2011). For these five features selected, the exact test indicates a significant difference (p-values are all less than 0.01) between two different pH samples.

As shown in Fig. 11B, Burkholderiaceae (selected by the four methods) was enriched in acidic condition (pH = 4.9). But when pH of soil was 8.5, its relative abundance was low. And from different pH samples, the relative abundance of Burkholderiaceae had a significant difference (p-value = 0.00058). Therefore, Burkholderiaceae could serve as marker to differentiate soil samples with different pH values. Previous work has reported that Burkholderiaceae needs oxalic acid as its source of carbon (Garrity, Bell & Lilburn, 2004), which partially support this finding.

Conclusions

The research in metagenomics becomes more and more popular as microbial communities were found to play important roles in many areas such as bioenergy, bioremediation and human health. The discovery of biomarker taxa for metagenomic datasets could facilitate identification of microbial community’s phenotype, thus making them important for community identification and even monitoring of the host or environment within which the community live.

However, current metagenomic datasets lack “ground truth” of biomarkers, making it hard for the assessment of computationally predicted metagenomic biomarkers by various methods. A properly generated synthetic dataset with embedded “ground truth” and taxonomical distribution properties similar to those of real metagenomic samples could make such assessment fair and easy. In this study, we have evaluated taxonomical distribution properties for different microbial communities, and found that their taxonomical distributions follow either normal distribution, gamma distribution, or the mixture of normal and gamma distribution. Therefore, in this work, synthetic datasets have been generated accordingly that could facilitate the assessment of biomarker identification methods.

We have proposed the MetaBoot method for metagenomic biomarker identification, which is a top-down method based on mRMR strategy and bootstrapping technique. The use of mRMR could reduce redundancies, while the use of bootstrapping could improve robustness of the MetaBoot method. It has been compared with two top-down methods (Metastats and LEfSe) and one bottom-up method (Wilcoxon rank-sum test) on simulated datasets, with results indicating that MetaBoot could identify more non-redundant biomarkers with high accuracy and robustness. On real oral and soil metagenomic datasets, it was also observed that MetaBoot could identify more reliable biomarkers for distinguish different types of microbial communities, showing that the results of MetaBoot were more biologically meaningful. Therefore, MetaBoot could serve well for metagenomic biomarker discovery.

Current taxonomical biomarker discovery methods still face several obstacles: Firstly most of them could identify biomarkers from only two groups of microbial communities, while biomarkers for a set of different groups could be more useful in several circumstances. Secondly, the biomarker sets (with multiple biomarkers) might be useful for complex samples such as microbial community, yet none has been done on how such sets could be optimized. Thirdly, with the advancement of whole genome sequencing, important functional biomarker identification using not only taxa but also genes would become feasible as well, yet current methods cannot identify functional biomarkers well. All these analytical bottlenecks will be addressed in the future development of MetaBoot and companion tools, and they in turn will help for better understanding of microbial communities and their impacts on our environment.

Supplemental Information

Supplemental Information 1 The supplemental files

This file includes four supplemental figures.

Click here for additional data file.

We thank Dr. Shi Huang for discussions about building MetaBoot, and Xingzhi Chang for comments about writing codes.

Additional Information and Declarations

Competing Interests

Author Contributions

All authors declare there are no competing interests.

Xiaojun Wang conceived and designed the experiments, performed the experiments, analyzed the data, contributed reagents/materials/analysis tools, wrote the paper, prepared figures and/or tables, reviewed drafts of the paper.

Xiaoquan Su and Xinping Cui contributed reagents/materials/analysis tools, reviewed drafts of the paper.

Kang Ning conceived and designed the experiments, performed the experiments, analyzed the data, contributed reagents/materials/analysis tools, wrote the paper, reviewed drafts of the paper.

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
