# Peer review of "MetaBoot: a machine learning framework of taxonomical biomarker discovery for different microbial communities based on metagenomic data"

_PeerJ, doi:10.7717/peerj.993_

## Round 0.1 · original submission · Major Revisions

It's nice to see idea to combine bootsrap and mRMR to find robust and non-redundant biomarkers. Both reviewers think the work is interesting and give some positive comments. Meanwhile, they also have some major concerns to be addressed. I suggest the authors to provide a point by point response letter and the revision.

For the removing of redundancy among selected features, an important reference is missing. The recent method in Nucl. Acids Res. (2013) 41 (4): e53 can beat mRMR in gene expression data.

Reviewer 1 ·

Basic reporting

none

Experimental design

none

Validity of the findings

none

Additional comments

none

Annotated reviews are not available for download in order to protect the identity of reviewers who chose to remain anonymous.

Reviewer 2 ·

Basic reporting

The authors give sufficient introduction and background of OTU-based microbial community analysis, but there exists a minor mistake. At the end of 4th paragraph, it's said that "none of these methods directly identify biological features responsible for group relationships". However, tools like Mothur does calculate beta-diversity of OTU abundance and find the OTUs (species-level) that have significantly different abundances.

Experimental design

In the "Classification accuracy analysis based on synthetic datasets", the authors need to clarify a key point of the experiment design. Was the MetaBoot feature selection carried out on the training set (50 samples) or on the training+testing sets (60 samples)? If it's done on 60 samples, the leaks of testing samples' information will lead to underestimation of classification error rate. In addition, 10 testing samples are not enough to get stable estimate of the classification accuracy. A better way is to run a 6-fold cross-validation.

Validity of the findings

For the soil dataset, the selected biomarkers are on phylum-level, thus it's impossible to understand their functional roles from biological / environmental point of view. It's better to use real datasets that can be depicted on genus-level. The authors may be interested in the following papers studying human gut microbiome.
(1) A human gut microbial gene catalogue established by metagenomic sequencing.
(2) Diet rapidly and reproducibly alters the human gut microbiome.

Additional comments

In general, the article was meaningful and well organized.

---

## Round 0.2 · accepted · Accept

The manuscript has been greatly improved after revision. Now I suggest its acceptance.